# TGTOD: A Global Temporal Graph Transformer for Outlier Detection at Scale

## Abstract

Graph outlier detection aims to identify anomalous substructures in graphs that deviate significantly from normal patterns. Traditional methods primarily focus on static graphs, overlooking the dynamic nature of real-world networks and ignoring valuable temporal signals crucial for outlier detection. While Transformers have revolutionized machine learning on time-series data, existing Transformers for temporal graphs face limitations in (1) restricted receptive fields, (2) overhead of subgraph extraction, and (3) suboptimal generalization capability beyond link prediction. In this paper, we propose TGTOD, a novel end-to-end Temporal Graph Transformer for Outlier Detection. TGTOD employs global attention to model both structural and temporal dependencies within temporal graphs. To tackle scalability, our approach divides large temporal graphs into spatiotemporal patches, which are then processed by a hierarchical Transformer architecture comprising Patch Transformer, Cluster Transformer, and Temporal Transformer. We evaluate TGTOD on three public datasets under two settings, comparing with a wide range of baselines. Our experimental results demonstrate the effectiveness of TGTOD, achieving AP improvement of 61% on Elliptic dataset. Furthermore, our efficiency evaluation shows that TGTOD reduces training time by 44× compared to existing Transformers for temporal graphs. To foster reproducibility, we make our implementation publicly available at https://anonymous.4open.science/r/tgtod.

## 1 Introduction

Outlier detection, a critical task in machine learning, aims to identify data points that significantly deviate from normal patterns. This task has become increasingly important in various domains, including fraud detection (Huang et al., 2022), anti-money laundering (Weber et al., 2019), and misinformation identification (Dou et al., 2021). Graph-structured data, which represents complex relationships as nodes connected by edges, has been widely adopted across numerous fields, such as citation networks (Zhao et al., 2020), social networks (Dou et al., 2021), and molecular structures (Wu et al., 2024a). Graph outlier detection focuses on identifying anomalous substructures within graphs. The inherent complexity of graph data, due to its non-Euclidean nature, makes this task particularly challenging. Traditionally, graph outlier detection methods have primarily focused on static graphs, where the structure and attributes remain constant over time. However, real-world graphs are often dynamic, evolving over time and providing rich temporal information that can be crucial for detecting outliers. Current static graph methods fall short when applied to temporal graphs, as they fail to capture the important temporal aspects necessary for effective outlier detection.

Recently, Transformers have revolutionized machine learning on language (Vaswani et al., 2017; Huang et al., 2024b) and vision (Dosovitskiy et al., 2020) with their powerful ability to model complex dependencies in data through self-attention mechanisms. Unlike traditional recurrent neural network architectures that rely heavily on sequential processing, Transformers utilize attention mechanisms to weigh the importance of different parts of the input data, allowing them to capture long-range dependencies effectively and efficiently. This self-attention mechanism in Transformers offers a promising direction by integrating temporal dynamics into the graph representation learning.

Although some recent efforts have adapted Transformers to temporal graphs, challenges remain. DyGFormer extracts one-hop interactions and feeds their neighbor, link, time, and co-occurrence

encoding into a Transformer to capture temporal edges between nodes (Yu et al., 2023). Similarly, SimpleDyG also models neighbors in temporal graphs as a sequence and introduces a temporal alignment technique to capture temporal evolution patterns (Wu et al., 2024b). Despite their potential, existing methods using Transformers on temporal graph face significant limitations.

- **Limited receptive field**: Current methods typically extract only *one-hop* neighboring subgraphs, restricting Transformers' receptive field and overlooking long-range spatiotemporal dependencies.
- **Training inefficiency**: These approaches often rely on subgraphs extraction for *each edge* (Hamilton et al., 2017), leading to significant computational overhead and limiting the training efficiency.
- **Task misalignment**: Existing Transformers are pretrained on *link prediction*, which may cause suboptimal generalization capability to node-level outlier detection due to mismatch of objectives.

To address these limitations, we propose TGTOD, a novel paradigm to apply Transformers on temporal graphs. (**Novelty 1**) We explore adopting *global spatiotemporal attention* on the entire temporal graph for node-level outlier detection. It allows TGTOD to capture not only the local spatiotemporal dependencies between temporal neighbors but also the global spatiotemporal dependencies across the entire graph and multiple timestamps. However, due to the quadratic complexity of query-key multiplication in the attention mechanism, direct global spatiotemporal attention is computationally infeasible for large-scale temporal graphs. (**Novelty 2**) Inspired by visual patching method used in video model (Brooks et al., 2024), we divide the large temporal graph into *spatiotemporal patches* to alleviate the challenge of scalability. For the spatial aspect, we use a graph clustering algorithm to partition the large graph into relatively small clusters. For the temporal aspect, we aggregate the interactions within each cluster into a snapshot with a specific time interval. These patches are then fed into a hierarchical Transformer, which includes Patch Transformer, Cluster Transformer, and Temporal Transformer. Through this approach, TGTOD significantly reduces the complexity of attention mechanisms while preserving global spatiotemporal receptive fields. (**Novelty 3**) Furthermore, unlike existing Transformers on temporal graphs pretrained on link prediction task, TGTOD is trained *end-to-end* on node-level outlier detection, ensuring better alignment with the downstream task and thus providing stronger generalization capability. Our contributions in this work are summarized as follows:

- **Global attention**: We propose TGTOD, making the first attempt to leverage global spatiotemporal attention for Transformers on temporal graphs for end-to-end node-level outlier detection.
- **Spatiotemporal patching**: We introduce a spatiotemporal patching method, which significantly improves the scalability of TGTOD and enables outlier detection on large-scale temporal graphs.
- **Evaluation**: Analysis and experiments demonstrate the effectiveness and efficiency of TGTOD.

The remainder of this paper is organized as follows: Section 2 provides a comprehensive review of related work in graph outlier detection, temporal graph learning, and graph Transformers. Section 3 presents a detailed description of our proposed methodology. Section 4 outlines our experimental setup, reports the results, and offers an in-depth analysis of our findings. Finally, Section 5 concludes the paper and discusses potential avenues for future research in this domain.

## 2 RELATED WORK

### 2.1 GRAPH OUTLIER DETECTION

Graph outlier detection is an essential task in machine learning, aimed at identifying anomalous structures within graphs. Due to its structural simplicity and broader applicability, most research focuses on node-level outlier detection, which can be extended to edge-level and graph-level tasks (Liu et al., 2024). A significant body of work focuses on unsupervised approaches, where outliers are detected based solely on the data without ground truth labels. For instance, DOMINANT (Ding et al., 2019) adopts a graph autoencoder (Kipf & Welling, 2016b) to reconstruct the input graph and identify anomalies based on the reconstruction error. CoLA (Liu et al., 2021b) detects anomalous nodes using contrastive learning (Ma et al., 2024). However, these unsupervised methods may fall short in scenarios requiring the identification of specific outliers with domain knowledge. In such cases, (semi-)supervised graph outlier detectors, which can learn from ground truth labels, are more suitable. For example, GAS (Li et al., 2019) employs an attention mechanism to detect spam reviews.

Studies like PCGNN (Liu et al., 2021a) conduct message passing in selected neighborhoods. Additionally, GATSep (Zimek et al., 2014) separates representation learning from anomaly detection. BWGNN (Tang et al., 2022) discerns both low-frequency and high-frequency signals, adaptively integrating them across various frequencies. GHRN (Gao et al., 2023) addresses heterophily from a graph spectrum perspective. Extensive works explore data augmentation (Liu et al., 2022a; 2023), active learning (Chang et al., 2024a), and fairness (Chang et al., 2024b) on graph outlier detection. Despite the proliferation of graph outlier detection methods, few consider the crucial aspect of time. In this work, we bridge this gap by capturing temporal information via Transformers.

## 2.2 TEMPORAL GRAPH LEARNING

Temporal graph learning has gained significant attention due to its ability to model dynamic relationships in real-world networks. For instance, EvolveGCN (Pareja et al., 2020) adapts the weights of graph neural networks over time using recurrent neural networks. TGN (Rossi et al., 2020) uses memory modules to capture long-term dependencies in temporal graphs. TGAT (Xu et al., 2020) adopts a self-attention mechanism and develops a functional time encoding technique. GraphMixer (Cong et al., 2023b) incorporates the fixed time encoding function into a link encoder to learn from temporal links. TGB (Huang et al., 2024a) provides a temporal graph benchmark for machine learning on temporal graphs. Despite these advancements, most existing temporal graph networks are trained on link level, e.g., for link prediction tasks. When the task is at the node level, e.g., node-level outlier detection, a node-level decoder is trained on the top of the frozen encoder. This two-stage training scheme may suffer from suboptimal generalization capability to node-level outlier detection. In addition, these methods require temporal subgraph extraction for each edge during training, leading to limited receptive fields and high computational overhead. Our work aims to address these limitations by leveraging the power of Transformers for temporal graph learning.

## 2.3 GRAPH TRANSFORMERS

Graph Transformers have emerged as a powerful approach for learning on graph data, combining the strengths of graph neural networks and Transformer architectures. These models aim to overcome limitations of traditional graph neural networks, such as over-smoothing and limited receptive fields, by leveraging the attention mechanisms of Transformers. Early work in this area includes Graphormer (Ying et al., 2021), which introduces centrality encoding, spatial encoding, and edge encoding to improve graph level representation. At the node level, NodeFormer (Wu et al., 2022) enables efficient computation via kernerlized Gumbel-Softmax, reduced the algorithmic complexity to linear. DIFFormer (Wu et al., 2023) proposes a graph Transformer with energy constrained diffusion. SGFormer (Wu et al., 2019) further improves the scalability of graph Transformers to handle large graphs more efficiently. CoBFormer (Xing et al., 2024) adopts bi-level architecture in global graph Transformers to alleviate over-globalization problem and improve the efficiency. These methods lay solid foundation for our work. Our work extends the global attention beyond node level to both spatial and temporal aspects on temporal graphs. Although some efforts have adapted Transformers to temporal graphs, existing methods inherit the problems of temporal graph network methods. SimpleDyG (Wu et al., 2024b) and DyGFormer (Yu et al., 2023) model one-hop temporal neighbors as a sequence and feed their encodings into Transformer to capture temporal edges. While these methods have shown promising results in temporal link prediction tasks, their application to node-level outlier detection remains limited. In this paper, we rethink the use of Transformers on temporal graphs and conduct global spatiotemporal attention on the entire temporal graph.

## 3 METHODOLOGY

Figure 1 provides an overview of the proposed TGTOD for *end-to-end* node-level outlier detection on temporal graphs. In TGTOD, we aim to conduct *global spatiotemporal attention* across the entire temporal graph. However, due to the quadratic complexity of attention mechanism, direct global spatiotemporal attention is computationally infeasible for large-scale temporal graphs. We thus propose to divide large-scale temporal graphs into relatively managable spatiotemporal patches to alleviate the challenge of scalability. Figure 1a shows a toy example of *spatiotemporal patching*. The obtained patches are then fed into a hierarchical Transformer architecture, as shown in Figure 1b, which includes Patch Transformer (PFormer), Cluster Transformer (CFormer), and Temporal Trans-

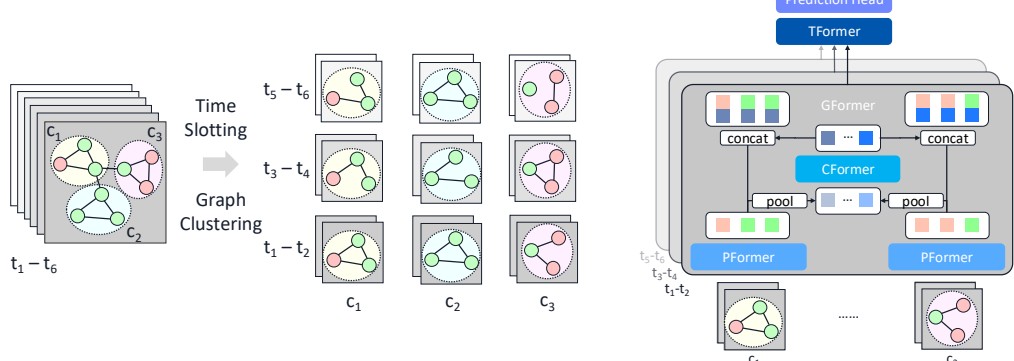

(a) Spatiotemporal patching method.

(b) Hierarchical architecture of TGTOD conducting global spatiotemporal attention.

Figure 1: An overview of TGTOD for end-to-end node-level outlier detection on temporal graphs.

former (TFormer). In this section, we start with the problem definition in Section 3.1 and introduce our spatiotemporal patching method in Section 3.2. Next, we define the hierarchical architecture of the proposed TGTOD in Section 3.3, followed by a detailed design of Patch Transformer, which further reduces computational complexity, in Section 3.4. We summarize the end-to-end training procedure of TGTOD in Section 3.5 and analyze the reduction in complexity achieved in TGTOD in Section 3.6. The notations used are summarized in Table 5 in Appendix A.

## 3.1 PROBLEM DEFINITION

Consider a temporal graph $\mathcal{G}$ consisting of a sequence of graph snapshots $\{\mathcal{G}^t\}_{t=1}^T$. In a snapshot $\mathcal{G}^t = (\mathcal{V}^t, \mathcal{E}^t, \boldsymbol{X}^t)$ at timestamp $t$, $\mathcal{V}^t$ represents the node set, $\mathcal{E}^t$ represents the edge set, and $\boldsymbol{X}^t$ is the feature matrix. $\mathcal{V} = \bigcup_{t=1}^T \mathcal{V}^t$ is the total node set of size $N$, while $\mathcal{E} = \bigcup_{t=1}^T \mathcal{E}^t$ is the total edge set. Given a partially labeled temporal graph $\mathcal{G}$, the problem of (semi-)supervised graph outlier detection is a binary classification task that learns a detector $f : v_i \in \mathcal{V} \rightarrow \{0, 1\}$, which classifies every node in $\mathcal{G}$ to either an inlier (0) or an outlier (1). In this paper, we consider two settings.

**Definition 1** *(Stationary Setting) In this setting, the node set and edge structure evolve over time, while both the features and the label of each node remain constant across all timestamps.*

This setting assumes that the nodes are always consistent over time. It is prevalent in real-world applications. A typical example is misinformation detection in social media. Although misinformation can be spread by different users over time, the main idea of misinformation (i.e., the feature) and the fact that it is misinformation (i.e., the label) remain unchanged.

**Definition 2** *(Non-Stationary Setting) In this setting, the node set and edge structure evolve over time, and both the features and the label of each node may also vary across different timestamps.*

This is a more general setting, allowing both the features and labels to evolve over time. An example is sensor network monitoring in the Internet of Things, where not only the devices in the network can be added or removed over time, but the device battery life and operational status (e.g., functioning or failure) of these devices can also change over time.

## 3.2 SPATIOTEMPORAL PATCHING

Our spatiotemporal patching approach is inspired by the video generation model Sora (Brooks et al., 2024), drawing parallels between video data and temporal graph data. A temporal graph is a sequence of graph snapshots with nodes, similar to how a video is a sequence of image frames with pixels. For instance, a 1-minute 1080p video at 24Hz consists of approximately 3 billion pixels. Directly treating these pixels as tokens and feeding them into Transformers, which have quadratic complexity, would result in prohibitive computational costs. In computer vision, this complexity

is managed by dividing the video data into visual patches, segmented over time and space. In this work, we adopt a similar strategy to divide large temporal graphs $\mathcal{G}$ into small spatiotemporal patches $\{p_c^s\}$, employing time slotting and graph clustering. Figure 1a illustrates an example of spatiotemporal patching. On the left, we have a complete temporal graph with 6 timestamps $t_1 - t_6$.

For **time slotting**, we aggregate timestamps over specific time intervals $\Delta t$ into time slots $\{s\}_{s=1}^S = \{[t, t + \Delta t]\}$, merging the node set $\mathcal{V}^s = \bigcup_{t=t_0}^{t_0+\Delta t} \mathcal{V}^t$ and the edge set $\mathcal{E}^s = \bigcup_{t=t_0}^{t_0+\Delta t} \mathcal{E}^t$. In the example shown in the figure, $\Delta t = 2$ and each slot contains 2 snapshots. The obtained slots are represented as rows on the right. To simplify the problem, we leave the removed nodes and newly added node in all timestamps as isolated nodes, and they do not affect results of outlier detection. The interval $\Delta t$ is a hyperparameter that is minimized within the constraints of available memory. A smaller $\Delta t$ enables more granular temporal processing, potentially capturing finer temporal dynamics at the cost of increased computational demands.

For **graph clustering**, we partition the aggregated large graph across all timestamps into relatively small and mutually exclusive clusters $\{\mathcal{V}_c\}_{c=1}^C$, where $\mathcal{V} = \bigcup_{c=1}^C \mathcal{V}_c$, using an efficient graph clustering algorithm METIS (Karypis & Kumar, 1998). This partitioning step is highly scalable and does not incur significant overhead as a preprocessing step. In the example shown in the figure, we partition the all nine nodes in the complete temporal graph into three closely connected clusters. Each cluster is listed in a column on the right. As a result, the complete temporal graph becomes $3 \times 3 = 9$ spatiotemporal patches, which are fed into the hierarchical Transformer architecture.

## 3.3 HIERARCHICAL TRANSFORMER ARCHITECTURE

The hierarchical architecture of TGTOD is illustrated in Figure 1b. The goal of TGTOD is to perform *global spatiotemporal attention* across the entire temporal graph. To achieve this with manageable computational complexity, we design a Transformer architecture that applies attention hierarchically. TGTOD first separates Spatial Transformer (GFormer) and Temporal Transformer (TFormer) to reduce the attention complexity. Spatial Transformer is further divided into Patch Transformer (PFormer, conducting attention within each patch) and Cluster Transformers (CFormer, handling inter-cluster attention across patches) to further reduce complexity. The output is then fed into Prediction Head tailored to specific settings and tasks–in our case, node-level outlier detection.

Within each patch, **Patch Transformer** (i.e., PFormer) conducts all pair attention within the patch to obtain the intra-patch node embedding for each node. $\boldsymbol{Z}_c^s = \text{PFormer}(\boldsymbol{X}_c^s)$, where $\boldsymbol{X}_c^s \in \mathbb{R}^{|\mathcal{V}_c| \times d}$ is the nodes feature matrix of patch $p_c^s$, and $\boldsymbol{Z}_c^s \in \mathbb{R}^{|\mathcal{V}_c| \times d'}$ is the intra-patch node embedding matrix. Here, $d$ and $d'$ are the feature dimension and the hidden dimension, respectively.

To extend the receptive field beyond individual patches, we employ an **Cluster Transformer** (i.e., CFormer). The intra-patch node embedding undergo pooling to produce the embedding of a patch: $\boldsymbol{p}_c^s = \text{pooling}(\boldsymbol{Z}_c^s)$, where $\boldsymbol{p}_c^s \in \mathbb{R}^{d'}$ is the embedding of patch $p_c^s$, and mean pooling is used in our implementation. The patch embeddings $\boldsymbol{P}^s = [\boldsymbol{p}_1^s, ..., \boldsymbol{p}_C^s]^\intercal$ are then processed by the Cluster Transformer to update $\bar{\boldsymbol{P}}^s = [\bar{\boldsymbol{p}}_1^s, ..., \bar{\boldsymbol{p}}_C^s]^\intercal$: $\bar{\boldsymbol{P}}^s = \text{CFormer}(\boldsymbol{P}^s)$, where $\bar{\boldsymbol{P}}^s, \boldsymbol{P}^s \in \mathbb{R}^{C \times d'}$.

When node $v_i$ in cluster $c$, the intra-patch node embedding $\boldsymbol{z}_i^s$ is concatenated with the corresponding undated patch embedding $\bar{\boldsymbol{p}}_c^s$ to form the spatial embedding $\bar{\boldsymbol{z}}_i^s = \text{concat}(\boldsymbol{z}_i^s, \bar{\boldsymbol{p}}_c^s)$, where $\bar{\boldsymbol{z}}_i^s \in \mathbb{R}^{2d'}$. To manage memory consumption efficiently, instead of updating all patch embeddings at once, we enable mini-batch training. We maintain an embedding table for each patch, and update the embedding of one patch $c$ at a time while keeping others frozen during each training step.

**Temporal Transformer** (i.e., TFormer) computes attention over spatial embeddings across time slots to obtain the final embedding of node $v_i$: $\tilde{\boldsymbol{Z}}_i = \text{TFormer}(\bar{\boldsymbol{Z}}_i)$, where $\bar{\boldsymbol{Z}}_i = [\bar{\boldsymbol{z}}_i^1, ..., \bar{\boldsymbol{z}}_i^S]^\intercal$ is the spatial embedding matrix of node $v_i$ across all $S$ time slots and $\tilde{\boldsymbol{Z}}_i, \bar{\boldsymbol{Z}}_i \in \mathbb{R}^{S \times 2d'}$.

To accommodate different tasks and settings, we adopt corresponding **Prediction Head** on the final embedding $\tilde{\boldsymbol{Z}}_i$. For stationary setting, $\tilde{\boldsymbol{Z}}_i$ is first pooled across the temporal dimension, either by mean or concatenation: $\hat{\boldsymbol{z}}_i = \text{pooling}(\tilde{\boldsymbol{Z}}_i)$, where $\hat{\boldsymbol{z}}_i \in \mathbb{R}^{2d'}$. Then, the pooled embedding is fed into a logistic regression model to estimate the outlier score $\hat{y}_i = \text{LogisticRegression}(\hat{\boldsymbol{z}}_i)$. For non-stationary setting, we directly adopt logistic regression on the final embedding $\tilde{\boldsymbol{z}}_i^s$ of node $v_i$ in time slot $s$ to estimate the outlier score $\hat{y}_i^s = \text{LogisticRegression}(\tilde{\boldsymbol{z}}_i^s)$, where $\tilde{\boldsymbol{Z}}_i = [\tilde{\boldsymbol{z}}_i^1, ..., \tilde{\boldsymbol{z}}_i^S]^\intercal$.

## 3.4 PATCH TRANSFORMER

For the detailed design of Transformers, we adopt vanilla Transformer for both Temporal Transformer and Cluster Transformer. For a detailed introduction to Transformer, please refer to Appendix B.1. However, we specifically design Patch Transformer to address two key limitations. (1) The quadratic complexity of vanilla Transformer prohibits the use of large patches, comprising the efficiency of TGTOD. Thus, we employ a *kernel method* (Wu et al., 2022) to approximate the all-pair attention, reducing complexity from quadratic to linear, thereby allowing the use of larger patch sizes. For more details, please refer to Appendix B.2. (2) Vanilla self-attention does not inherently account for graph structure, limiting its ability to capture structural information. To address this, we integrate a Graph Neural Network (GNN) as a *residual connection* within the Patch Transformer. The output is formulated as a weighted summation of the GNN output and the Transformer output:

$$\text{PFormer}(\boldsymbol{X}_c^s) = \alpha \cdot \text{GNN}(\boldsymbol{X}_c^s, \mathcal{E}_c^s) + (1 - \alpha) \cdot \text{AppxAttn}(\boldsymbol{X}_c^s), \tag{1}$$

where $\alpha$ is a hyperparameter graph weight. We adopt a Graph Convolutional Networks (GCN) (Kipf & Welling, 2016a) as the GNN implementation.

## 3.5 END-TO-END OUTLIER DETECTION TRAINING

Existing methods (Yu et al., 2023) on temporal graphs handle node-level tasks (e.g., node classification) in a two-stage manner. The temporal graph encoder is first pretrained on link prediction task in an unsupervised manner (i.e., without node labels) to learn a general embedding for each node, and a decoder specifically designed for node-level task is trained on the top of the obtained embedding in a supervised manner (i.e., with node labels). This two-stage training pipeline separates the representation learning from the specific task, which may result in suboptimal generalization capability to node-level outlier detection due to the mismatch in objectives.

Different from the above methods, TGTOD can be trained end-to-end for node-level tasks, e.g., node-level outlier detection. It enables *direct optimization* of the outlier detector objective during training, learning a more discriminative node representation specifically tailored to the task of node-level outlier detection. Moreover, this end-to-end training paradigm in TGTOD avoids the need for separate pretraining and downstream training stages, leading to a *more efficient training process*. In **stationary** setting, the output of TGTOD $\hat{y}_i$ is used to estimate loss values with the binary cross-entropy loss function for end-to-end training:

$$\mathcal{L} = \frac{1}{N} \sum_i^N y_i \log \hat{y}_i + (1 - y_i) \log(1 - \hat{y}_i). \tag{2}$$

Similary, in **non-stationary** setting, the output of TGTOD $\hat{y}_i^s$ is used:

$$\mathcal{L} = \frac{1}{N} \sum_i^N \sum_s^S y_i^s \log \hat{y}_i^s + (1 - y_i^s) \log(1 - \hat{y}_i^s). \tag{3}$$

This methodology ensures that TGTOD effectively captures both spatial and temporal dependencies in temporal graphs, enhancing scalability and generalization for graph outlier detection.

## 3.6 COMPLEXITY ANALYSIS

TGTOD aims to perform global spatiotemporal attention across the entire temporal graph while maintaining manageable computational complexity. Here, we analyze the reduction in computational complexity achieved by each component of TGTOD. In the original attention mechanism, the multiplication of query and key matrices leads to quadratic complexity. Considering each node at each timestamp as a token, we have $NT$ tokens in total, where $N$ is the number of nodes and $T$ is the number of timestamps. The complexity of direct global spatiotemporal attention is thus $\Omega(Attn) = N^2 T^2$.

By separating Spatial Transformer and Temporal Transformer, we prioritize the more important and denser attention across time within patches in the same position, while omitting the less critical and sparser attention between patches in different positions and time slots. This reduces the complexity

to $N^2 + T^2$. For Spatial Transformer, we partition the graph with $N$ nodes into $C$ clusters, where the average cluster size is $M = N/C$. Separating PFormer and CFormer further reduces the complexity to $M^2 + C^2 + T^2$. Additionally, the approximation in Patch Transformer reduces its complexity from quadratic to linear, resulting in a final complexity of $M + C^2 + T^2$.

To illustrate the significance of this reduction, consider a temporal graph with $N = 10^6$ nodes, $T = 1000$ timestamps, and $C = 1000$ clusters. The complexity of direct global spatiotemporal attention would be $1,000,000,000,000,000,000$, while the complexity of TGTOD is reduced to $2,001,000$, representing a substantial improvement in computational efficiency.

## 4 EXPERIMENTS

To evaluate our method and compare it with contemporary approaches, we conduct experiments in a unified environment. In this section, we first introduce the experimental setup in Section 4.1, followed by performance evaluation in Section 4.2. Next, we conduct a hyperparameter analysis in Section 4.3. Finally, we empirically assess the efficiency of the methods in Section 4.4.

### 4.1 EXPERIMENTAL SETUP

**Datasets**. Table 1 provides the statistics of the three datasets used in our experiments. In the table, #Nodes stands for the number of nodes, and #Edges represents for the number of edges. #Features denotes the raw feature dimension. Outlier means the outlier ratio in the ground truth label. #Time is the number of timestamps, i.e., the number of graph snapshots. The Stationary column indicates whether the dataset is in stationary setting, where the node features and labels remain unchanged across timestamps. For Elliptic and DGraph, due to the nature of the datasets, we can only consider the stationary setting. For FiGraph, we consider both stationary (using the lastest features and labels) and non-stationary settings. Detailed descriptions for each dataset are available in Appendix C.

Table 1: Statistics of datasets.

| Dataset | #Nodes | #Edges | #Features | Outlier | #Time | Stationary |
|---------|--------|--------|-----------|---------|-------|------------|
| Elliptic | 203,769 | 234,355 | 165 | 9.8% | 49 | ✓ |
| DGraph | 3,700,550 | 4,300,999 | 17 | 1.3% | 821 | ✓ |
| FiGraph | 236,692 | 873,346 | 247 | 2.8% | 9 | ✗ |

**Metrics**. We follow the existing literature in graph outlier detection (Liu et al., 2022b; Tang et al., 2024) to evaluate the outlier detection performance with three commonly used metrics: AUC, AP, and Recall@k. In addition, we evaluate the efficiency with the number of parameters, training time, and memory consumption. The detailed description of each metric is available in Appendix D.

**Baselines**. To evaluate the performance of proposed TGTOD, we compare it with a wide range of baselines. We first compare TGTOD with general graph neural networks (GNN), including SGC (Wu et al., 2019), GCN (Kipf & Welling, 2016a), GraphSAGE (Hamilton et al., 2017), GAT (Veličković et al., 2018), and GIN (Xu et al., 2018). We also compare TGTOD with temporal graph networks (Temporal): TGN (Rossi et al., 2020), TGAT (Xu et al., 2020), and GraphMixer (Cong et al., 2023a). In addition, we include state-of-the-art graph outlier detectors (Detector) in the comparison, including GAS (Li et al., 2019), PCGNN (Liu et al., 2021a), GATSep (Zhu et al., 2020), BWGNN (Tang et al., 2022), and GHRN (Gao et al., 2023). Since Transformers for temporal graphs is still a relatively new research area, limited methods are available for comparison. We thus only compare TGTOD with two Transformer-based methods (Transformer), static Graph Transformer (GT) (Shi et al., 2020) and DyGFormer (Yu et al., 2023). For the ablation study, we also implement the variants of TGTOD by removing one of the components in TGTOD, including w/o TFormer, w/o CFormer, w/o PFormer, and w/o GNN.

**Implementation Details**. For detailed experimental implementation, we modify GADBench (Tang et al., 2024) and DyGLib (Yu et al., 2023) to benchmark graph outlier detection performance. To reduce the influence of randomness, most experiments are repeated 10 times, with results reported as the average and standard deviation. However, due to excessive runtime of temporal graph methods in DyGLib–specifically TGN, TGAT, GraphMixer, and DyGFormer–these experiments are only run 3 times. Additional details can be found in Appendix E.

Table 2: Detection AUC, AP, and Recall@k (%) on **Elliptic** and **DGraph** under **stationary** settings.

| Category | Method | Elliptic | | | DGraph | | |
|---|---|---|---|---|---|---|---|
| | | AUC | AP | Rec@k | AUC | AP | Rec@k |
| GNN | SGC | 75.4±1.2 | 12.8±0.7 | 11.0±1.7 | 66.1±0.3 | 2.4±0.1 | 4.2±0.2 |
| | GCN | 81.4±1.8 | 21.9±2.9 | 25.0±6.0 | 75.9±0.2 | 4.0±0.1 | 7.1±0.2 |
| | GraphSAGE | 85.3±0.7 | 32.9±6.0 | 37.3±4.6 | 75.6±0.2 | 3.8±0.1 | 7.0±0.4 |
| | GAT | 84.9±1.9 | 25.2±5.6 | 27.9±11.3 | 75.9±0.2 | 3.9±0.1 | 7.4±0.2 |
| | GIN | 82.7±2.0 | 23.5±4.9 | 27.3±8.3 | 74.0±0.2 | 3.3±0.1 | 5.9±0.2 |
| Temporal | TGN | 82.8±2.7 | 37.6±2.4 | 39.9±5.9 | OOM | OOM | OOM |
| | TGAT | 85.4±1.6 | 36.5±8.8 | 41.7±6.2 | 70.7±0.1 | 2.8±0.1 | 4.4±0.1 |
| | GraphMixer | 86.8±1.1 | 39.9±8.6 | 40.8±3.7 | 73.7±0.4 | 3.0±0.0 | 4.8±0.1 |
| Detector | GAS | 85.6±1.6 | 27.9±6.6 | 34.6±9.6 | 76.0±0.2 | 3.8±0.1 | 6.8±0.2 |
| | PCGNN | 85.8±1.8 | 35.6±10.2 | 40.4±12.0 | 72.0±0.3 | 2.8±0.0 | 5.0±0.2 |
| | GATSep | 86.0±1.4 | 26.4±4.5 | 31.3±8.8 | 76.0±0.2 | 3.9±0.1 | 7.5±0.3 |
| | BWGNN | 85.2±1.1 | 26.0±3.5 | 31.7±6.2 | 76.3±0.1 | 4.0±0.1 | 7.5±0.3 |
| | GHRN | 85.4±1.9 | 27.7±6.6 | 33.3±10.3 | 76.1±0.1 | 4.0±0.1 | 7.5±0.2 |
| Transformer | GT | 85.1±1.5 | 25.1±4.5 | 26.3±11.1 | 75.8±0.1 | 3.9±0.1 | 7.5±0.2 |
| | DyGFormer | 79.8 ± 2.3 | 21.3 ± 6.3 | 22.8 ± 5.5 | 70.3±0.1 | 2.8±0.1 | 4.8±0.1 |
| Ours | w/o TFormer | 87.4±0.9 | 57.3±3.9 | 57.3±3.2 | 76.4±0.3 | 3.6±0.1 | 5.8±0.4 |
| | w/o CFormer | 88.7±1.0 | 60.8±5.0 | 60.8±1.7 | 78.0±0.4 | **4.1±0.1** | **6.5±0.4** |
| | w/o PFormer | 88.3±1.4 | 59.6±7.0 | **60.9±5.9** | 77.6±0.0 | 3.9±0.1 | 6.0±0.4 |
| | w/o GNN | 87.8±0.8 | 49.3±5.2 | 52.2±3.8 | 72.5±0.2 | 2.8±0.1 | 4.4±0.3 |
| | TGTOD | **89.2±0.5** | **64.4±5.9** | 60.7±2.6 | **78.3±0.3** | **4.1±0.1** | **6.5±0.4** |

Table 3: Detection AUC, AP, and Recall@k (%) on **FiGraph** under different settings.

| Category | Method | Stationary | | | Non-Stationary | | |
|---|---|---|---|---|---|---|---|
| | | AUC | AP | Rec@k | AUC | AP | Rec@k |
| GNN | SGC | 48.9±3.7 | 3.6±0.9 | 1.2±2.5 | 64.0±1.3 | 5.2±0.4 | 9.8±1.3 |
| | GCN | 53.2±3.1 | 5.3±1.3 | 9.4±3.1 | 70.8±1.2 | 7.1±0.3 | 12.3±1.1 |
| | GraphSAGE | 60.7±10.4 | 7.3±3.5 | 10.6±7.4 | 60.4±6.9 | 4.4±1.5 | 5.1±3.6 |
| | GAT | 74.6±2.3 | 11.8±3.0 | 14.4±7.9 | 80.5±1.0 | 11.4±1.8 | 15.0±4.1 |
| | GIN | 67.3±8.1 | 8.8±3.0 | 13.8±6.1 | 72.7±5.8 | 8.1±2.2 | 11.6±3.1 |
| Temporal | TGN | 62.8±5.1 | 6.4±1.8 | 7.4±3.2 | 77.8±0.2 | 10.4±1.1 | 16.3±1.4 |
| | TGAT | 72.9±7.2 | 11.8±8.1 | 14.8±11.6 | 78.5±1.6 | 11.1±0.6 | 16.5±1.7 |
| | GraphMixer | 61.6±10.7 | 8.7±3.6 | 11.1±5.6 | 79.7±0.4 | 12.7±0.3 | 17.7±0.9 |
| Detector | GAS | 72.8±4.3 | 8.3±1.0 | 5.6±3.4 | 80.1±0.4 | 12.0±0.6 | 17.8±1.7 |
| | PCGNN | 76.5±2.3 | 14.9±2.5 | 16.2±7.5 | 77.2±0.9 | 8.3±0.4 | 11.2±1.4 |
| | GATSep | 75.9±2.7 | 12.9±1.6 | 14.4±4.0 | 77.0±4.2 | 10.7±2.1 | 15.0±3.5 |
| | BWGNN | 77.6±2.8 | 13.3±2.0 | 16.2±4.1 | 80.4±1.1 | 11.8±1.8 | 16.5±2.6 |
| | GHRN | 78.2±2.1 | 13.5±2.2 | 13.8±2.5 | 79.8±1.7 | 11.2±2.0 | 14.9±4.3 |
| Transformer | GT | 74.1±4.9 | 10.8±2.7 | 11.9±4.4 | 80.4±1.3 | 12.1±1.2 | 16.3±2.5 |
| | DyGFormer | 60.5±4.9 | 4.6±0.5 | 1.9±3.2 | 69.2±2.7 | 5.2±1.1 | 6.2±3.4 |
| Ours | w/o TFormer | 77.0±3.6 | 12.6±2.5 | 12.5±6.2 | 76.1±3.2 | 12.5±3.0 | 12.5±5.6 |
| | w/o CFormer | 78.1±5.9 | 15.0±2.9 | 14.4±6.3 | 78.0±2.1 | 13.5±1.7 | 13.8±4.7 |
| | w/o PFormer | 79.1±2.2 | 14.6±3.9 | 15.0±8.0 | 80.4±2.1 | 15.0±3.8 | 15.6±4.2 |
| | w/o GNN | **80.2±2.9** | 12.7±2.3 | 13.8±4.7 | 80.1±3.3 | 13.6±1.3 | 16.9±4.0 |
| | TGTOD | 78.6±2.3 | **16.0±3.0** | **17.2±5.8** | **80.6±2.5** | **15.1±2.4** | **18.8±4.8** |

## 4.2 EVALUATION ON PERFORMANCE

In this section, we analyze the outlier detection performance of TGTOD compared to a wide range of baselines across three datasets under two settings. Tabel 2 presents the outlier detection performance results under stationary setting in AUC, AP, and Recall@k for different types of baselines and TGTOD on Elliptic and DGraph datasets. Table 3 shows the results on FiGraph dataset under both stationary and non-stationary settings. In each table, we highlight the best performance of our method in **bold**, and underline the best performance achieved by other methods. "OOM" indicates that the method is out of memory during training.

By comparing TGTOD with baselines and variants across three datasets under both stationary and non-stationary settings, we have the following observations:

- TGTOD demonstrates strong effectiveness, outperforming the best baselines on most metrics on three of the datasets. Notably, on Elliptic dataset, TGTOD achieves 64.4 in AP, significantly surpassing the best baseline GraphMixer by 61%, which records an AP of 39.9.
- In the ablation study, removing individual components (e.g., TFormer, CFormer, PFormer, and GNN) from TGTOD typically results in worse performance compared to the complete model. This underscores the importance of global spatiotemporal attention in TGTOD.
- Existing temporal graph methods underperform on DGraph and FiGraph due to suboptimal generalization from link prediction. DyGFormer is particularly ineffective, likely due to its sensitivity to hyperparameter settings. These results prove the superiority of end-to-end training of TGTOD.

### 4.3 HYPERPARAMETER ANALYSIS

In this section, we conduct experiments to analyze the impact of hyperparameters on the performance of TGTOD. For most of the hyperparameters, e.g., number of layers, we follow the default settings in the previous papers. Our focus is on the graph weight $\alpha$ in Equation 1 for hyperparameter analysis. We experiment with different values of $\alpha$ on Elliptic dataset, and the results are presented in Figure 2. We observe that TGTOD achieves the best performance in all three metrics when $\alpha = 0.5$, indicating that both the GNN output and the Transformer output in Equation 1 are crucial for its effectiveness. This result highlights the importance of the specifically designed Patch Transformer, validating its necessity for optimal performance.

### 4.4 EFFICIENCY ANALYSIS

Efficiency is another important aspect for the application of TGTOD on large-scale temporal graphs. We empirically evaluate the efficiency of existing Transformer-based method for temporal graphs, DyGFormer, along with our proposed method, TGTOD, on the largest dataset, DGraph. The evaluation results, summarized in Table 4, detail three key metrics: #Param (the number of model parameters), Time (training time per epoch), and Memory (main memory consumption during training). Further details regarding the metrics are available in Appendix D.2. Our results show that TGTOD significantly outperforms DyGFormer in model size, training duration, and memory usage. Notably, TGTOD accelerates training by 44×, highlighting the efficiency of spatiotemporal patching.

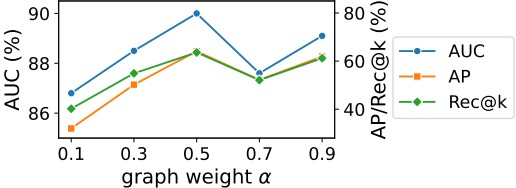

Table 4: Efficiency evaluation of temporal graph Transformers on DGraph dataset.

| Model | #Param | Time (s) | Memory |
|---|---|---|---|
| DyGFormer | 982,659 | 1,942 | 49G |
| TGTOD | 6,865 | 44 | 16G |

Figure 2: Hyperparameter analysis of $\alpha$ (graph weight) in Equation 1 on Elliptic dataset.

### 5 CONCLUSION AND FUTURE WORK

By rethinking the application of temporal graph Transformers for outlier detection, in this study, we present TGTOD, making the first attempt to leverage global spatiotemporal attention for outlier detection in temporal graphs. Our method significantly improves scalability through spatiotemporal patching while preserving a global receptive field, enabling effective end-to-end outlier detection on large-scale temporal graphs. Through comprehensive analysis and experiments on real-world datasets, we demonstrate that TGTOD not only outperforms existing state-of-the-art methods in detection performance but also exhibits superior computational efficiency.

This work establishes a new benchmark in temporal graph outlier detection and opens up promising avenues for future research. Potential further explorations include extending TGTOD and spatiotemporal patching to broader temporal graph learning tasks beyond outlier detection, as well as pretraining it as a foundational model for various downstream tasks. We believe these research directions will foster the development of more effective and scalable algorithms capable of managing the complex spatiotemporal dynamics inherent in real-world graph data.

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

# A    NOTATIONS

In this section, we summarize the notations used in this paper.

Table 5: Summary of notations.

| Symbol | Description |
|---|---|
| $\mathcal{G}, \mathcal{V}, \mathcal{E}$ | Entire temporal graph, and its node set and edge set |
| $\mathcal{G}^t, \mathcal{V}^t, \mathcal{E}^t, \boldsymbol{X}^t$ | Graph snapshot at timestamp $t$, and its node set, edge set, and feature matrix |
| $\mathcal{V}^s, \mathcal{E}^s$ | Node set and edge set of timeslot $s$ |
| $\mathcal{V}_c$ | Node set of cluster $c$ |
| $N, T, \Delta t$ | Number of nodes, number of timestamps, and time interval for time slotting |
| $f, v_i$ | Outlier detector function, the $i$th node in the graph |
| $p_c^s$ | Spatiotemporal patch for cluster $c$ at slot $s$ |
| $\boldsymbol{X}_c^s$ | Patch feature matrix for nodes in cluster $c$ at slot $s$ |
| $\boldsymbol{z}_i^s, \boldsymbol{Z}_c^s$ | Intra-patch node embedding for node $i$ at slot $s$, and its matrix |
| $\boldsymbol{p}_c^s, \boldsymbol{P}^s$ | Patch embedding of cluster $c$ at slot $s$, and their matrixs |
| $\bar{\boldsymbol{p}}_c^s, \bar{\boldsymbol{P}}^s$ | Updated patch embedding of cluster $c$ at slot $s$, and their matrix |
| $\bar{z}_i^s, \bar{\boldsymbol{Z}}_i$ | Spatial embedding of node $i$ at slot $s$, and its matrix |
| $\tilde{z}_i^s, \tilde{\boldsymbol{Z}}_i$ | Updated final embedding for node $i$ at slot $s$, and its matrix |
| $\hat{\boldsymbol{z}}_i$ | Pooled final embedding of node $i$ |
| $\hat{y}_i, \hat{y}_i^s$ | Estimated outlier score for node $i$ (at slot $s$) |
| $y_i, y_i^s$ | Ground-truth label for node $i$ (at slot $s$) |
| $d, d'$ | Feature dimension, hidden dimension |
| $\alpha$ | Hyperparameter of graph weight in PFormer |

# B    ATTENTION MECHANISM IN TRANSFORMERS

In this section, we introduce two types of attention mechanisms used in TGTOD, including vanilla attention in Section B.1 and an attention approximation with linear complexity in Section B.2.

## B.1    VANILLA ATTENTION

The attention mechanism is a core component of Transformers, introduced by (Vaswani et al., 2017). Its primary purpose is to enable the model to focus on different parts of the input sequence. The attention mechanism models the dependencies across the entire sequence regardless of their distance, which is essential for handling long-range dependencies in sequence data.

The Transformer utilizes a self-attention mechanism, also known as scaled dot-product attention, which computes attention scores between every pair of tokens in the input sequence. The inputs to the attention mechanism are three matrices: Queries ($Q$), Keys ($K$), and Values ($V$), all of which are linear projections of the input embeddings. These projections are defined as:

$$Q = XW_Q, \quad K = XW_K, \quad V = XW_V \tag{4}$$

where $X \in \mathbb{R}^{n \times d}$ is the input sequence of $n$ tokens, each with dimension $d$, and $W_Q, W_K, W_V \in \mathbb{R}^{d \times d_k}$ are learned projection matrices for the queries, keys, and values, respectively, with $d_k$ denoting the dimensionality of the queries and keys.

The attention weights are computed as the scaled dot product of the queries and keys:

$$\text{Attn}(X) = \text{softmax}\left(\frac{QK^\intercal}{\sqrt{d_k}}\right) V = \text{softmax}\left(\frac{XW_Q(XW_K)^\intercal}{\sqrt{d_k}}\right) XW_V \tag{5}$$

where $\frac{1}{\sqrt{d_k}}$ is a scaling factor. The softmax function is applied to the dot products to normalize the attention weights, ensuring they sum to 1. The weights indicate how much focus each query should place on each value, enabling the model to weigh different parts of the input sequence differently.

This attention mechanism is extended to Multi-Head Attention (MHA). Rather than computing a single set of attention scores, the model projects the queries, keys, and values into $h$ different subspaces, each of dimensionality $d_k$, and computes independent attention heads:

$$\text{MHA}(Q, K, V) = \text{concat}(\text{head}_1, \dots, \text{head}_h)W_O, \quad \text{head}_i = \text{Attn}(QW_{Q_i}, KW_{K_i}, VW_{V_i}) \quad (6)$$

and $W_O \in \mathbb{R}^{hd_k \times d}$ is the output projection matrix. The multi-head mechanism allows the model to attend to different parts of the input sequence from different representational subspaces, thereby enhancing its capacity to capture diverse dependencies.

Despite the powerness of attention mechanism, its quadratic complexity of query-key multiplication in Equation 5 limits the scalability of the model. In Appendix B.2, we will introduce an attention approximation to reduce the complexity to linear.

## B.2 LINEAR ATTENTION APPROXIMATION

Wu et al. (2022) introduces a novel approach to reduce the computational complexity of attention on large graphs, reducing the complexity of attention mechanism from quadratic to linear. Specifically, a *kernelized Gumbel-Softmax operator* is employed to approximate the all-pair attention. The key idea is to transform the dot-product attention into a kernel function, which can then be efficiently approximated using random feature maps. In this attention mechanism, the message-passing at node level can be expressed as:

$$z_u^{(l+1)} = \sum_{v=1}^{N} \kappa \left( W_Q^{(l)} z_u^{(l)}, W_K^{(l)} z_v^{(l)} \right) \cdot W_V^{(l)} z_v^{(l)}, \quad (7)$$

where $W_Q^{(l)}$, $W_K^{(l)}$, and $W_V^{(l)}$ are the learnable parameters for the queries, keys, and values, respectively, in the $l$-th layer, and $\kappa(\cdot, \cdot)$ is a pairwise similarity function (i.e., the kernel function). Instead of computing the full dot-product attention, we approximates the kernel function $\kappa(a, b)$ using random features map as proposed by Rahimi & Recht (2007). The approximation is given by:

$$\kappa(a, b) \approx \langle \phi(a), \phi(b) \rangle, \quad (8)$$

where $\phi(\cdot)$ is a random feature map that approximates the kernel. The dot product between $\phi(a)$ and $\phi(b)$ is significantly faster to compute than the original dot-then-exponentiate operation, reducing the overall complexity from $O(N^2)$ to $O(N)$.

## C DETAILED DESCRIPTIONS OF DATASETS

**DGraph** (Huang et al., 2022): DGraph is a large-scale graph dataset provided by Finvolution Group, including around 3 million nodes, 4 million dynamic edges, and 1 million node labels. The nodes represent user accounts within a financial organization that offers personal loan services, with edges indicating that one account has designated the other as an emergency contact. Nodes labeled as fraud correspond to users exhibiting delinquent financial behavior. For accounts with borrowing records, outliers are accounts with a history of overdue payments, while inliers are those without such a history. The dataset includes 17 node features derived from user profile information.

**Elliptic** (Weber et al., 2019): This graph dataset contains over 200,000 Bitcoin transaction nodes, 234,000 directed payment flow edges, and 165 dimensional node features. The dataset maps Bitcoin transactions to real-world entities, categorizing them into both licit categories, including exchanges, wallet providers, miners, and legal services, and illicit categories, such as scams, malware, terrorist organizations, ransomware, and Ponzi schemes.

**FiGraph** (Wang, 2024): This dataset presents a temporagph graph for financial anomaly detection. Spanning from 2014 to 2022, FiGraph captures the dynamics of financial interactions through 9 distinct temporal snapshots. The graph consists of 730,408 nodes and 1,040,997 edges. It has five types of nodes and four types of edges. The dataset focus on target nodes with features, while the background nodes without features provide structure information for anomaly detection.

## D  DETAILED DESCRIPTIONS OF METRICS

In this section, we provide a detailed description of the metrics we used in this paper for the evaluation of both effectiveness and efficiency.

### D.1  EFFECTIVENESS METRICS

**AUC**: area under receiver operating characteristic curve, which is constructed by plotting the true positive rate against the false positive rate across varied determined threshold levels. An AUC of 1 indicates perfect prediction, while an AUC of 0.5 suggests the model cannot distinguish between classes. AUC is preferable to accuracy for evaluating outlier detection tasks because it is not affected by imbalanced class distributions.

**AP**: average precision, also known as area under precision-recall curve, summarizes the precision-recall curve by calculating the weighted mean of precision values at each threshold, where the weight corresponds to the increase in recall from the previous threshold. As a metric that balances both recall and precision. In most outlier detection applications, FPR and FNR are both important.

**Rec@k**: Outliers are typically rare compared to the large number of normal samples, but they are the primary focus in outlier detection. We propose to use Recall@k to assess how effectively detectors rank outliers relative to normal samples. Here, k is set to the number of outliers in labels. Recall@k is the number of true outliers among the top-k samples in the ranking list, divided by k.

### D.2  EFFICIENCY METRICS

**Number of parameters**: the total count of learnable parameters in the model. This metric provides insight into the model's complexity. A lower parameter count indicates a more efficient model. In implementation, we count the number of parameters with `numel()` method for each `torch.nn.Module.parameters()` in the model.

**Training time**: the time required to train the model for one epoch using the maximum batch size that fits within the 40GB GPU memory constraints of NVIDIA A100. For methods employing two-stage training, the total training time is calculated as the sum of the training times for both stages.

**Memory consumption**: the peak main memory usage of the method during training on the given dataset, measured when using CPU only. This metric provides insight into the method's memory efficiency, which is particularly important for large-scale graph processing tasks.

## E  IMPLEMENTATION DETAILS

**Hardware**. All of our experiments were performed on a Linux server with an AMD EPYC 7763 64-core CPU, 256GB RAM, and an NVIDIA A100 GPU with 40GB memory.

**Dependencies**. The key libraries and their versions used in experiments are as follows: Python 3.10, CUDA 11.8, PyTorch 2.1.0 (Paszke et al., 2019), PyG 2.5.3 (Fey & Lenssen, 2019), DGL 2.4.0 (Wang, 2019), and PyGOD 1.1.0 (Liu et al., 2024).

**Hyperparameters**. For baselines implemented by GADBench and DyGLib, we directly adopt the default hyperparameters in the original library with minor modifications to fit our experimental environment. TGTOD is mostly implemented with default hyperparameters following previous works. We only customize a few hyperparameters presented in Table 6.

Table 6: Hyperparameters of TGTOD on different datasets.

|  | Elliptic | DGraph | FiGraph |
| --- | --- | --- | --- |
| time slotting interval $\Delta t$ | 1 | 10 | 1 |
| number of clusters $C$ | 64 | 64 | 1 |
| graph weight $\alpha$ | 0.8 | 0.8 | 0.9 |
| hidden dimension $d'$ | 32 | 16 | 16 |

