# OpenReview forum: "TGTOD: A Global Temporal Graph Transformer for Outlier Detection at Scale"
_ICLR.cc/2025/Conference — ICLR 2025 Conference Withdrawn Submission_

### Official Review · Reviewer_yQ4P · 2024-10-24

**Soundness:** 2
**Presentation:** 2
**Contribution:** 2
**Rating:** 5
**Confidence:** 5

**Summary:**

The authors address the problem of anomaly detection over temporal graphs, a relatively less explored area compared to anomaly detection on static graphs. They highlight limitations in learning temporal signals using Transformers for this task.

Based on these limitations, the authors propose an end-to-end Temporal Graph Transformer for Outlier Detection (TGTOD). TGTOD improves scalability by dividing large temporal graphs into spatiotemporal patches, followed by three Transformer networks to model both structural and temporal dependencies in temporal graphs. The experimental results demonstrate the effectiveness of TGTOD against leading baselines in outlier detection tasks.

**Strengths:**

+ The proposed method is simple, effective, and scalable.

+ The experimental results show overall improvement over baselines.

+ Code for reproducing the experiments is provided.

**Weaknesses:**

+ The focus on Transformers for temporal graph learning raises concerns about novelty, as similar approaches have been extensively explored.

+ The experiments are not fully convincing. Important datasets, baselines, and ablation studies are missing (see detailed comments below).

+ Some claims and illustrations are vague and require more clarity (see detailed comments below).

**Questions:**

+ Why is SimpleDyG mentioned in the related work but missing from the comparative analysis?

+ Since the primary focus is on outlier detection, I suggest including some static outlier detection methods for comparison, instead of relying solely on common GNNs like GCN and SGC.

+ The use of only three datasets is insufficient. Common benchmarks for temporal outlier detection, such as Wikipedia, Reddit, and Mooc[1], are notably missing from the experiments.

+ The definitions in lines 192-193 are inaccurate. Generally, node labels are dynamically changing and are usually defined with a timestamp $t$.

+ Some state-of-the-art baselines are missing, such as SAD[2] and SLADE[3].

+ The claim that “existing Transformers are pretrained on link prediction…” is not entirely correct. Many temporal Transformers (e.g., TGAT, DyGFormer, SimpleDyG) are trained in an end-to-end manner for node- or link-level tasks.

+ In Table 4, TGTOD shows good efficiency over DyGFormer. However, DyGFormer was not designed to be an efficient method for temporal graph learning. The authors should include more relevant baselines like SimpleDyG and TGAT for a comprehensive comparison.

+ Ablation studies on varying time slots and the number of clusters are missing.

+ In Table 6, the time slot is set to 1 for most datasets, which is a common setting in temporal graph learning. What is the necessity of the “patching” step in this context?



[1] JODIE: Predicting Dynamic Embedding Trajectory in Temporal Interaction Networks. KDD 2019.

[2] SAD: Semi-Supervised Anomaly Detection on Dynamic Graphs. IJCAI 2023

[3] SLADE: Detecting Dynamic Anomalies in Edge Streams without Labels via Self-Supervised Learning. KDD 2024.

---

### Official Review · Reviewer_z1SG · 2024-10-29

**Soundness:** 1
**Presentation:** 2
**Contribution:** 2
**Rating:** 3
**Confidence:** 5

**Summary:**

This paper introduces TGTOD, a new end-to-end Temporal Graph Transformer designed for Outlier Detection in dynamic graphs.

**Strengths:**

1. Outlier detection in dynamic graphs is an important problem.
2. Given the limited number of existing models for outlier detection in dynamic graphs, this paper makes a valuable contribution by focusing on this direction and proposing a new method specifically for outlier detection in dynamic graphs.

**Weaknesses:**

1. Unclear Motivation: The motivation behind this work is not well-founded. For example, the authors mention "Limited receptive field" as a motivation; however, neither DyGFormer nor SimpleDyG was specifically designed for outlier detection. The use of first-order neighbors is a deliberate design choice to avoid aggregating irrelevant information, which has proven effective in link prediction tasks. Thus, this choice is not inherently a limitation of receptive field. Additionally, the concept of "task misalignment" seems misplaced since previous models were not intended for outlier detection, making "pretraining" irrelevant in this context.

2. Poor Organization: The paper dedicates substantial space to background knowledge and related works, yet fails to incorporate these works in the experimental comparisons. This organizational choice limits the paper’s coherence and weakens its argument for contribution.

3. Limited Experiments: The experimental section is insufficient to convincingly demonstrate the model’s efficacy. Although several related works (e.g., NodeFormer, DIFFormer, SGFormer, CoBFormer) are discussed, none are included in the experimental comparisons. Furthermore, the baselines used (e.g., GCN, GraphSage) are basic, while more advanced temporal models like CAWN and TCL would be more appropriate. The limited metrics (AP and AUC) are inadequate for evaluating performance on an imbalanced dataset with a low anomaly rate; metrics such as F1-score would provide a more complete evaluation. The absence of ablation studies and hyperparameter analysis further detracts from the experimental rigor.

4. Limited Novelty: The novelty of the model is minimal, as it merely combines three existing transformer architectures without any modification, contributing little innovation in terms of model design.

**Questions:**

Please refer to the weakness

---

### Official Review · Reviewer_xFxm · 2024-11-04

**Soundness:** 2
**Presentation:** 3
**Contribution:** 2
**Rating:** 3
**Confidence:** 5

**Summary:**

This paper focuses on the challenge of graph outlier detection in temporal graphs. The authors argue that existing transformer-based models are inadequate for temporal graphs due to their quadratic computational cost and suboptimal generalization capabilities. To overcome these limitations, they propose partitioning the given graph into multiple subgraphs and applying hierarchical transformers to these subgraphs. Their method, TGTOD, integrates both graph neural networks and transformers to effectively capture structural and temporal dependencies within the temporal graph. Experimental results demonstrate the superior performance of TGTOD on three real-world temporal graphs, outperforming general graph neural networks, graph transformers, and graph outlier detectors.

**Strengths:**

**S1**: The paper studies the problem of graph outlier detection by focusing on temporal graphs. This problem is important and has many practical applications in real-world scenarios.

**S2**: The authors conduct extensive and thorough experiments to demonstrate the effectiveness of their proposed transformer framework across three real-world datasets.

**Weaknesses:**

**W1**: The primary concern regarding this work centers on its substantial lack of novel insights and originality in the proposed framework. The core components of the proposed framwork appear to be largely derivative of existing approaches, with minimal innovative additions. Firstly, the idea of graph partitioning as a strategy for reducing computational complexity, while effective, cannot be considered a novel contribution, as this approach has been extensively explored and implemented in existing models like ClusterGCN [R1]. Secondly, both the temporal transformer and cluster transformer essentially replicating the vanilla transformer architecture without substantial modifications or improvements tailored to graph-specific challenges. Similarly, the patch transformer component appears to be a direct adaptation of NodeFormer [R2]. Thirdly, integrating different components through weighted summation of GNN and transformer outputs has been previously introduced in SGFormer [R3].

**W2**: The time complexity analysis is cursory and lacks rigor. It omits crucial considerations regarding the complexity of the METIS clustering algorithm, and the presentation lacks formal asymptotic notations. Additionally, the numerical examples provided are overly simplified, neglecting critical constant terms that could significantly impact real-world performance, such as the number of clusters, hidden dimensions, and attention head counts. A more rigorous analysis should encompass these factors and present complexity bounds with appropriate asymptotic notation.

**W3**: The efficiency analysis is insufficient. The authors only compare their proposed TGTOD with DyGFormer, which does not offer a comprehensive assessment of its efficiency. It is imperative to include comparisons against a wider array of state-of-the-art methods and other baseline models for a more thorough evaluation.

**W4**: The authors claim that existing transformer-based models suffer from restricted receptive fields. However, transformers are renowned for their ability to leverage a global receptive field, which is a significant advantage over traditional graph neural networks. As such, transformers can effectively address the constraints imposed by graph structures and capture long-range dependencies. This statement requires further justification and clarification to be convincing.

---

[R1] W. Chiang, X. Liu, S. Si, Y. Li, S. Bengio and C. Hsieh. Cluster-GCN: An Efficient Algorithm for Training Deep and Large Graph Convolutional Networks. 2019. SIGKDD: 257–266.

[R2] Q. Wu, W. Zhao, Z. Li, D. Wipf and J. Yan. NodeFormer: A Scalable Graph Structure Learning Transformer for Node Classification. 2022. NeurIPS(35): 27387-27401.

[R3] Q. Wu, W. Zhao, C. Yang, H. Zhang, F. Nie, H. Jiang, Y. Bian and J. Yan. SGFormer: Simplifying and Empowering Transformers for Large-Graph Representations. 2023. NeurIPS(36): 64753-64773.

**Questions:**

**Q1**: How does the graph partitioning approach in TGTOD differ from that used in ClusterGCN? Additionally, how does TGTOD's focus on temporal graphs influence its partitioning strategy compared to the static graph approach of ClusterGCN?

**Q2**: Can existing scalable node-level anomaly detection methods, such as XGBGraph [R4], be directly applied to address the challenges of temporal outlier detection? If not, what specific modifications or adaptations are necessary to ensure these methods effectively handle the dynamic nature of temporal graphs? If they can be applied directly, how does TGTOD compare with XGBGraph in terms of effectiveness and efficiency when dealing with temporal outlier detection?

**Q3**: It appears that the authors may have omitted necessary parentheses in the loss function presented in Equations 2 and 3.

**Q4**: To provide a comprehensive efficiency analysis of TGTOD, it would be helpful to report the results of other baseline models.

---

[R4] J. Tang, , F. Hua, Z. Gao, P. Zhao and J. Li. GADBench: Revisiting and Benchmarking Supervised Graph Anomaly Detection. 2023. NeurIPS(36): 29628-29653.

---

### Official Review · Reviewer_uCmf · 2024-11-08

**Soundness:** 3
**Presentation:** 3
**Contribution:** 2
**Rating:** 3
**Confidence:** 5

**Summary:**

This paper studies how to use Transformer for outlier detection in temporal graphs at scale. A temporal graph transformer with hierarchical architecture is proposed to handle partitioned temporal graph patches with improved scalability. The proposed TGTOD is evaluated on three datasets and outperforms standard Transformer baselines in both performance and computational efficiency.

**Strengths:**

- The hierarchical Transformer structure, combined with spatiotemporal patching is a promising approach to improving scalability.

- TGTOD performs in the evaluation, validating the feasibility of using patch-based methods for financial and fraud detections in temporal graphs.

**Weaknesses:**

- Partitioning large graphs into clusters is a well-established technique for dealing with scalability issues, e.g., ClusterGCN, GraphSAINT.
- Current model designs (e.g., choice of clustering algorithm, patch size, and hierarchy) lack clear, evidence-based justification.
- Results appear to be highly tailored to specific datasets for outlier detection, while the broader applicability of TGTOD to other temporal graph domains or for general purpose of spatio-temproal graph learning remains uncertain.

Chiang, Wei-Lin, et al. "Cluster-gcn: An efficient algorithm for training deep and large graph convolutional networks." KDD'19.
Zeng, Hanqing, et al. "Graphsaint: Graph sampling based inductive learning method." ICLR'20

**Questions:**

- Q1 Could the authors provide some insights on the choice of clustering algorithm and patching interval? Specifically, the choice to use METIS for clustering is not directly tied to empirical or theoretical benefits specific to TGTOD’s design.

- Q2 How does the partitioning of the temporal graph affect spatio-temporal correlation?

- Q3 Have the authors tried directly using an efficient Trasnformer (e.g. Nodeformer) with single global-attention but not patching?

- Q4 Could the authors provide a more clear comparison between TGTOD and Nodeformer, since they share the same kernelized message passing with GNN embedded? Is FiGraph that used C=1 cluster (Table 6) corresponding to this case?

- Q5 How does TGTOD’s scalability compare to non-Transformer-based methods, such as GNNs?

Wu, Qitian, et al. "Nodeformer: A scalable graph structure learning transformer for node classification." NeurIPS'22

---

### Note · Authors · 2024-11-28

I have read and agree with the venue's withdrawal policy on behalf of myself and my co-authors.